# The Effect of Aspirin Use on Incident Hepatocellular Carcinoma—An Updated Systematic Review and Meta-Analysis

**DOI:** 10.3390/cancers15133518

**Published:** 2023-07-06

**Authors:** Jonathan Abdelmalak, Natassia Tan, Danny Con, Guy Eslick, Ammar Majeed, William Kemp, Stuart K. Roberts

**Affiliations:** 1Department of Gastroenterology, Alfred Health, Melbourne, VIC 3004, Australia; j.abdelmalak@alfred.org.au (J.A.); n.tan@alfred.org.au (N.T.); a.majeed@alfred.org.au (A.M.); w.kemp@alfred.org.au (W.K.); 2Department of Medicine, Central Clinical School, Monash University, Melbourne, VIC 3145, Australia; 3Department of Gastroenterology, Austin Health, Heidelberg, VIC 3084, Australia; dannycon302@gmail.com; 4Clinical Links Using Evidence-Based Data (CLUED) Pty. Ltd., Sydney, NSW 2060, Australia; guy.eslick@sydney.edu.au

**Keywords:** aspirin, liver cancer, hepatocellular carcinoma, incidence, risk, meta-analysis, systematic review

## Abstract

**Simple Summary:**

Aspirin has been observed to reduce the risk of developing hepatocellular carcinoma (HCC). This review pooled together the results of all available published studies and found that aspirin reduces the risk of HCC by around 30%. In patients with cirrhosis, this effect was not seen and, overall, patients treated with aspirin were at a higher risk of bleeding, as expected. Our findings provide compelling observational evidence for the use of aspirin as a potential preventative treatment for patients at risk of HCC, but highlight the need for further study into the optimal patient group that would benefit, and the need to balance against the risk of bleeding.

**Abstract:**

An increasing number of observational studies have described an association between aspirin use and a reduced risk of incident hepatocellular carcinoma. We performed this meta-analysis to provide a comprehensive and updated aggregate assessment of the effect of aspirin on HCC incidence. Two independent authors performed a systematic search of the literature, utilising the Medline, Embase, Scopus, and PubMed databases. A total of 16 studies (12 cohort studies, and 4 case-control studies) were selected for inclusion, with a large number of studies excluded, due to an overlapping study population. The pooled analysis of cohort studies involving a total population of approximately 2.5 million subjects, 822,680 aspirin users, and 20,626 HCC cases demonstrated a 30% reduced risk of HCC associated with aspirin use (adjusted HR 0.70, 95%CI 0.60–0.81). There was a similar but non-significant association observed across the case-control studies (adjusted OR 0.60, 95%CI 0.32–1.15, *p* = 0.13), which involved a total of 1961 HCC cases. In a subgroup meta-analysis of patients with cirrhosis, the relationship between aspirin use and incident HCC diminished to non-significance (adjusted HR 0.96, 95%CI 0.84–1.09). Aspirin use was associated with a statistically significant increase in bleeding events when all relevant studies were pooled together (adjusted HR 1.11, 95%CI 1.02–1.22). Prospectively collected data should be sought, to define the optimal patient group in which aspirin is safe and effective for the chemoprophylaxis of HCC.

## 1. Introduction

Hepatocellular carcinoma (HCC) is a major and rapidly increasing cause of global premature morbidity and death, projected to account for 60% of all deaths due to chronic liver disease by 2030 [1]. Increasing in incidence at a significantly greater rate than all other malignancies [2], HCC is now one of the most common global causes of cancer-related deaths [3], with an estimated yearly burden of 830,000 deaths in 2020 [4]. Key factors underpinning these phenomena are the burgeoning epidemic of obesity, as well as chronic viral hepatitis, and alcohol, via the causation of chronic liver disease and liver cancer. Indeed, up to 90% of patients with HCC have underlying cirrhosis [5], with the incidence of HCC in cirrhosis being 2–4% per annum [6]. Despite advances in treatment, the 5-year survival rate of HCC remains poor, at approximately 20% [7,8], with around 90% of all individuals who develop HCC dying of their disease [4,6]. This highlights the urgent need for effective preventative strategies. Currently, there is no specific chemopreventative agent in use for HCC; however, HCC screening in high-risk patients has been shown to be effective in reducing the overall mortality [9,10].

Aspirin is an inexpensive, widely available drug that targets the cyclo-oxygenase group of enzymes. It is thought, based on animal studies, to interfere with HCC carcinogenesis, through both its anti-platelet [11,12] and anti-inflammatory [13,14] effects. The role of aspirin in the primary prevention of colorectal cancer has been definitively established [15,16,17]; however, its utility in the prevention of other cancers remains subject to investigation. Several observational studies have been published to date, attempting to assess the effect of aspirin use on incident HCC. The results have varied; however, recently, a number of large high-quality studies have suggested that there is a duration-dependent effect of aspirin in reducing the incidence of HCC, supporting a causal relationship. A number of meta-analyses have been published [18,19,20,21,22,23,24,25,26,27,28,29,30]; however, many of these have erroneously included multiple studies drawing from the same population registries, calling into question the validity of their results. In this systematic review and meta-analysis, we aimed to pool all the available observational data, and perform an accurate meta-analysis, to provide an updated assessment of the relationship between aspirin use and the risk of incident HCC.

## 2. Materials and Methods

### 2.1. Information Sources and Search Strategy

The systematic review and meta-analysis were performed according to the Preferred Reporting Items for Systematic Reviews and Meta-analyses (PRISMA) and Meta-analysis Of Observational Studies in Epidemiology (MOOSE) guidelines [31,32]. The protocol has not been registered. An independent systematic search of the literature was conducted by two authors (JA and NT), utilising the MEDLINE, Embase, Scopus, and PubMed databases, with each database searched from its inception up to the 17 May 2023. The search terms used in each database were tailored to the search software, and are specifically outlined in Appendix B. The reference lists of relevant manuscripts were also screened, to assess potential studies missed from the systematic search results; however, no such additional studies were identified.

### 2.2. Eligibility Criteria

The criteria for inclusion in the systematic review and meta-analysis were: (1) the full manuscript was available in English; (2) it reported a quantitative statistic describing the association between incident HCC or liver cancer and either aspirin alone, or non-steroidal anti-inflammatory or antiplatelet drugs, with aspirin as a subpopulation; (3) it reported on a unique study population, with no included persons across more than one study.

### 2.3. Selection Process

Two authors then, after excluding duplicates, independently reviewed the titles and abstracts of each result, to determine its eligibility for inclusion. The full text of relevant studies was then independently reviewed, to confirm that it met the eligibility criteria. In cases of multiple studies drawing from the same population registry, the years of data extraction and the exclusion criteria were noted. Studies were carefully chosen, with preference given to the highest-quality study, as assessed by the modified Newcastle–Ottawa Quality Assessment Scale (mNOS) [33] and, if there was equal quality, the study or studies involving the largest number of subjects possible was selected. The remaining studies were excluded.

### 2.4. Data Collection and Items and Bias Assessment

The following data were collected from the included studies: the first author, the year of publication, the country of study, the type of population, the exclusion criteria where relevant, the specific definition of aspirin use, the specific criteria used for the HCC diagnosis, the crude risk estimate statistic, the adjusted risk estimate statistic if available, the variables used for the adjustment of the model, and the quality of each study using the mNOS. Additionally, further data collected for the cohort studies were: the total number of participants, the number of aspirin users, the number of controls, and the number of HCC cases in the aspirin users and controls. For case-control studies, the number of cases and controls was tabulated together with the rates of aspirin use in each group. Subgroup analyses, dose/duration analyses, and bleeding risk estimates were recorded, where available.

### 2.5. Statistical Analysis

The pooled hazard ratios (HR) and odds ratios (OR), and 95% confidence intervals (CI) were calculated, using a random-effects meta-analysis to account for between-study variability. We quantified the degree of heterogeneity between studies using the *I*^2^ statistic, which represents the percentage of the total variability between studies, where we considered 0–25% to be low, 25–75% to be moderate, and >75% to be high. The homogeneity between studies was tested using the chi-square test. The main effect for each analysis was tested using the z-test. The primary analysis incorporated adjusted effect sizes where available. A sensitivity analysis was conducted by performing a meta-analysis on only the unadjusted effect sizes. A two-sided *p* < 0.05 was considered to be statistically significant. All analyses were performed using Stata/IC 16.1 (StataCorp LLC, College Station, TX, USA, 2020).

## 3. Results

### 3.1. Study Selection

The initial screening identified a total of 4132 studies. Of these, 4083 were excluded after the screening of titles and abstracts, as they failed to meet the inclusion criteria. After a review of the full texts of the remaining 49 studies, a further 13 studies were found to fail to meet the inclusion criteria, and were excluded. A large number of cohort studies involved the same population, in part due to the availability of large national databases, such as the Taiwan National Health Insurance Research Database, the South Korea National Health Insurance Service Database, and the Hong Kong Hospital Authority Clinical Data repository, and also due to the duplication of efforts in the same longitudinal US cohorts and European population registries. A total of 20 studies were excluded due to overlapping study populations (eight in South Korea, seven in Taiwan, two in the USA, two in Hong Kong, one in Sweden), and the details of these are outlined in Table 1. There was finally a total of sixteen studies, twelve cohort [34,35,36,37,38,39,40,41,42,43,44,45] and four case-control [46,47,48,49], selected for inclusion. The overall search and study selection process is summarised in Figure 1.

### 3.2. Study Characteristics and Risk of Bias

A summary of the selected studies and their characteristics, including bias assessment, is presented in Table 2 and Table 3 (cohort and case-control studies, respectively). Across all of the cohort studies, a total of more than 2,404,876 individual participants were included, involving 822,680 aspirin users, and 20,626 distinct cases of HCC. The four case-control studies involved a total of 1961 cases of HCC, and 11,681 controls. These studies were published between the years 2000 and 2023. Seven studies were based in East Asia, five in North America, and the remaining four in Europe. Across the sixteen studies, eight focused specifically on populations with an elevated risk of HCC, with either chronic viral hepatitis, non-alcoholic fatty liver disease, or cirrhosis, while the remaining eight included broad cross-sections of the general population. A total of 14 out of the 16 studies were defined as high-quality studies (mNOS 7 or greater).

### 3.3. Association between Aspirin Use and Risk of Incident HCC

#### 3.3.1. Cohort Studies

The meta-analysis of the twelve cohort studies demonstrated that aspirin users, when compared with non-users, had a 30% lower risk of developing HCC (adjusted HR 0.70, 95% CI 0.60–0.81, *p* < 0.01), with a high degree of statistical heterogeneity (*I*^2^ = 87.37%, *p* < 0.01) (Figure 2). The sensitivity analysis using only unadjusted crude data demonstrated a similar result, with HR 0.65, 95% CI 0.45–0.94, *p* = 0.02 with similarly significant statistical heterogeneity (*I*^2^ = 84.28%, *p* < 0.01) (Figure 3).

#### 3.3.2. Case-Control Studies

The meta-analysis of the case-control studies demonstrated a considerable degree of statistical heterogeneity (*I*^2^ = 87.01%, *p* < 0.01), with a trend towards risk reduction that was non-significant, with a wide confidence interval (adjusted OR 0.60, 95% CI 0.32–1.15, *p* = 0.13) (Figure 4). There was a minimal difference in the results utilising only unadjusted crude data (OR 0.72, 95% CI 0.38–1.35, *p* = 0.31) (Appendix A).

#### 3.3.3. Cirrhosis

A total of five cohort studies reported on patients with a diagnosis of cirrhosis specifically, involving a large number of patients (n = 122,645). Two studies [37,39] (n = 6768) performed a subgroup analysis for patients with hepatitis B-related cirrhosis, while the other three [34,38,44] (n = 115,877) wholly comprised patients with cirrhosis. One study encompassed the breadth of the aetiology of cirrhosis [44], one focused on patients with only alcohol-related cirrhosis [38], and the final study involved patients with solely HBV- and HCV-related cirrhosis [34]. In patients with cirrhosis, aspirin use appeared to have a non-significant effect on HCC incidence (adjusted HR 0.96, 95% CI 0.84–1.09, *p* = 0.52), without significant statistical heterogeneity (*I*^2^ = 0%, *p* = 0.05) (Figure 5). The sensitivity analysis using only unadjusted crude data also demonstrated a non-significant relationship, albeit with a trend towards risk reduction (HR 0.62, 95% CI 0.31–1.26, *p* = 0.19) (Appendix A).

### 3.4. Dose and Duration Dependent Effects

A total of four cohort studies [39,41,42,43], and one case-control [49] study reported the risk of incident HCC across different levels of aspirin exposure, in order to establish if there was a dose/duration dependent effect. All five of these studies stratified their aspirin users into users of different set durations of therapy [39,41,42,43,49]. One study [42] also reported on the daily dose and frequency of aspirin. Four of the five studies [39,41,42,43] that looked at set durations of aspirin therapy indicated a duration-dependent effect, whilst one [49] showed no significant difference. Notably, the single study [42] that reported on aspirin daily dose found that only low-dose aspirin (<163 mg/day) led to a statistically significant HCC risk reduction (adjusted HR 0.39, 95% CI 0.17–0.91), in contrast to high-dose aspirin (adjusted HR 0.67, 95% CI 0.42–1.08). Furthermore, this study found that aspirin taken less than once a day was associated with a similar HCC risk reduction to aspirin taken daily.

### 3.5. Bleeding Risk

There were four cohort studies, involving 391,046 subjects, including 36,698 aspirin users, that reported on the association between aspirin use and bleeding risk [37,39,43,44]. All four studies focused on a different definition of bleeding, with three of the four studies only including gastrointestinal bleeding events. One study included all cases of gastrointestinal bleeding [43], one study (with a population comprised wholly of patients with cirrhosis) focused only on the incidence of variceal haemorrhage [44], and another reported only cases of peptic ulcer bleeding [39]. The fourth and largest study included all cases of major bleeding, defined as an intracranial haemorrhage or a gastrointestinal bleed requiring hospital admission or transfusion [37]. The meta-analysis of three of the studies [37,43,44] demonstrated that aspirin use was associated with a modest but statistically significant increase in the risk of bleeding (adjusted HR 1.11, 95%CI 1.02–1.22, *p* = 0.02), without significant statistical heterogeneity (*I*^2^ = 0%, *p* = 0.54) (Figure 6). The overall result of an approximate 10% increase in the risk of bleeding was similarly found in the remaining study [39], which only reported the crude incidence rates of bleeding in aspirin users and non-users (6.13% cumulative 5-year incidence, and 5.52% cumulative 5-year incidence, respectively).

## 4. Discussion

Globally, hepatocellular carcinoma remains a major and increasing cause of cancer-related death [3], with the vast majority of patients with HCC dying of their disease [70,71]. Preventative strategies are therefore needed as therefore needed, to reduce the individual patient’s risk of hepatocellular carcinoma incidence and mortality.

Aspirin is widely used in the primary and secondary prevention of cardiovascular disease; however, its anti-cancer effects (primarily through its anti-cyclooxygenase activity) have been widely described, including in liver cancer mouse models [11,14]. It is now recommended for consideration as a preventative treatment in colorectal cancer, with a broad evidence base of observational and randomised controlled trials supporting its efficacy [15,72,73,74,75,76,77,78]. To date, there have been a large number of observational studies reporting a negative association between aspirin use and HCC, but no randomized controlled trials have been performed yet.

Multiple meta-analyses [18,19,20,21,22,23,24,25,26,27,28,29,30] have been published over the last few years, summarising the observational data in this area, with the majority of these concluding that aspirin does have a significant association with a reduced HCC risk. Unfortunately, the majority of these systematic reviews have introduced bias into their risk estimates via the erroneous inclusion of multiple studies drawing from the same population. By mistakenly including the same subjects and HCC cases more than once, the estimate of precision of these analyses will have been overestimated and, depending on which populations were duplicated, the overall risk estimate may have been significantly swayed. Due to concern for the unreliability of previous meta-analyses on this basis, as well as the interval publication of further high-quality studies [37,38], we sought to perform an updated and methodologically sound meta-analysis, in order to provide the most accurate and updated insights into the effect of aspirin use on HCC incidence.

Similar to previous reports, our updated systematic review and careful meta-analysis affirms the presence of a negative relationship between aspirin use and incident HCC. However, unlike previous reports, our meta-analysis is the first to report a diminished risk-reduction effect in cirrhosis, and is also the first to observe, in this setting, a statistically significant increase in bleeding events associated with aspirin. Our review involves 16 studies across 8 countries, with just under 2.5 million participants, and more than 22,000 unique cases of HCC, whilst acknowledging and excluding a further 20 otherwise-eligible studies, due to overlapping study populations.

We found that the evidence of risk reduction in HCC incidence with aspirin was similar, regardless of the type of study evaluated. The overall reduction in the HCC risk was 30% among cohort studies, and 40% across case-control studies, although the confidence in the risk estimate from the case-control studies was poor, due to a wide confidence interval, driven by significant heterogeneity and a relatively small number of studies. Furthermore, the sensitivity analyses utilising unadjusted data demonstrated similar results, increasing the confidence in the strength and direction of relationship observed.

Of the five studies that examined differences between the varying duration of aspirin use, four of them provided supportive evidence of a duration-dependent effect [39,41,42,43]. Two previous meta-analyses evaluating the data across these studies have provided further support for the existence of an exposure-dependent effect [23,26]. While these data together lend significant weight to the assertion that aspirin is the causative factor in the HCC risk reduction observed in this meta-analysis, randomised controlled trial data are ultimately required to confirm the causal nature of the observed relationship.

Interestingly, the subgroup analysis of over 120,000 patients with cirrhosis demonstrated a comparatively diminished effect of aspirin use on incident HCC, with a statistically insignificant trend towards risk reduction in both the primary and sensitivity analyses. Given that the postulated mechanism of aspirin’s chemoprophylactic role in hepatocarcinogenesis operates primarily through a reduction in the hepatic accumulation of inflammatory cells, and a reduction in the hepatocellular injury that drives oncogenic mutations [12], it is plausible that this effect would have a diminished impact in the already-cirrhotic population, and maximal impact in earlier stages of liver disease. Only two studies [37,39] directly compared patients with and without cirrhosis, both in the chronic-hepatitis-B cohort. Both of these studies found a strong significant relationship in the non-cirrhotic cohort, and a non-significant relationship in cirrhosis, in keeping with the results of our meta-analysis. Further work is needed in order to best define the subpopulation that would maximally benefit from aspirin therapy for HCC chemoprevention, but our results highlight that the non-cirrhotic population, particularly those with chronic hepatitis B, may receive the greatest benefit.

The expected increase in bleeding risk associated with aspirin therapy is a major consideration to balance against the reduction of HCC risk, particularly in patients with underlying liver disease, who are at higher risk of both variceal and non-variceal gastrointestinal bleeding. Previous reports from the general population have suggested that aspirin leads to a modest increase in the risk of major bleeding (RR estimates between 1.40 and 1.43) [79,80,81]. All four included studies observed a non-significant trend towards increased bleeding in aspirin users; however, on pooling these, we revealed a statistically significant 11% increase in bleeding associated with aspirin use. It is likely that the true increase in risk is even higher than this, as with these retrospective studies, there is a high likelihood of a selection bias in which patients with otherwise-low individual bleeding risk were continued on aspirin, while others with higher risks of bleeding had aspirin discontinued.

Given that our data are solely observational, reverse causality should be considered as a possible contributor to the observed results. Patients with more advanced liver disease, where the risk of HCC is higher, may be less likely to be treated with aspirin therapy, due to concern about the bleeding risk. Additionally, it is important to consider the possibility of competing risks (in particular, cardiovascular events with associated non-liver-related mortality) biasing the aspirin-user group to an artificially lower rate of observed incident HCC, and magnifying the apparent risk reduction. It should also be noted that aspirin use might be a surrogate marker for engagement with healthcare, and be positively associated with other factors known to reduce HCC risk, such as the moderation of alcohol intake, metabolic risk factor modification, and adherence with antiviral treatments for hepatitis B and C infection. Furthermore, patients taking aspirin therapy are more likely to be on statin or metformin therapy, and these medications have also been shown to be associated with HCC risk reduction [82,83].

The significant degree of heterogeneity we observed in this meta-analysis may be due to bias, stemming from one of several reasons. Firstly, and most importantly, the majority of studies failed to consider and/or adjust for key related variables in their risk estimate analysis. Three of the twelve cohort studies [35,38,45], and two of the four case-control studies [46,47] did not provide an adjusted risk model at all. A further cohort study [36], and the remaining two case-control studies [48,71] did not account for cirrhosis status, which is the single most important risk factor for HCC, and potentially a subgroup in which there is a lessened impact of aspirin therapy. The status of the oncogenic hepatitis B virus was even less well documented and adjusted for, with none of the case-control studies, and only five out of the twelve cohort studies, accounting for hepatitis B status in their adjusted risk estimates (three studies [37,39,43] by adjusting for antiviral use; two studies [40,41] by excluding all chronic hepatitis B cases). It is certainly possible for adherence to aspirin therapy to be associated with adherence to antiviral medication, thereby confounding the results in the studies that failed to control for this key factor. Statin and metformin use, which have both been reported to negatively attenuate the incidence of HCC development, were also rarely examined, with only four out of twelve cohort studies [36,39,41,44], and one case-control study [49], adjusting for the use of both of these medications in their multivariate risk estimates. Similarly, there was generally poor documentation of, and adjustment for, multiple other key factors known to affect HCC risk, such as alcohol use, the presence of NAFLD/metabolic risk factors, a family history of HCC, and hepatitis C and hepatitis delta viral status. A second likely source of heterogeneity was the variability in the definition and capture of aspirin use. There was a wide spectrum of what constituted an ‘aspirin user’, with some studies accepting a minimum total of 30 doses during the study period, and others requiring daily use for periods of 3, 6, or 12 months, and yet others accepting any reported aspirin use, even a single dose. It should also be noted that one case-control study [46] included non-aspirin NSAIDs, along with aspirin, as their relevant exposure in assessing HCC risk compared to baseline. Systematic differences were also evident in how aspirin use was captured. The majority of Asian and Swedish studies relied on prescription records to capture aspirin use, with the remainder relying on patient-reported aspirin use, through health questionnaires. For the former, this raises the possibility of misclassification bias, with users of non-prescription over-the-counter aspirin being defined as non-users. For the latter, there is a possibility of recall bias confounding the results. Finally, the reliance on ICD coding for the definition of HCC in some studies [35,46,49] could have contributed to the observed heterogeneity, via the erroneous inclusion of intrahepatic cholangiocarcinoma in the HCC risk estimates associated with aspirin use.

## 5. Conclusions

This updated systematic review and careful meta-analysis suggests that aspirin use is associated with a strong duration-dependent effect in reducing the risk of incident HCC, and that this relationship appears to be somewhat diminished in cirrhosis. Aspirin use, as expected, is associated with a small increase in bleeding risk. There is therefore an urgent need to perform randomised placebo-controlled trials in well-phenotyped populations, such as those with non-cirrhotic chronic hepatitis B infection, to establish the clinical efficacy and safety of aspirin use for HCC chemoprophylaxis.

## Figures and Tables

**Figure 1 cancers-15-03518-f001:**
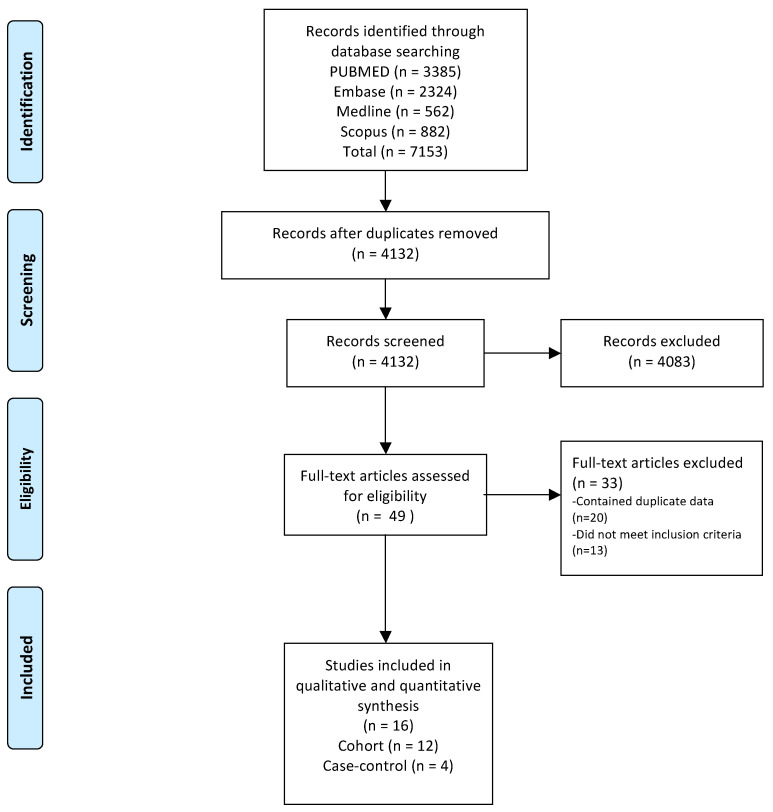
The search flow diagram.

**Figure 2 cancers-15-03518-f002:**
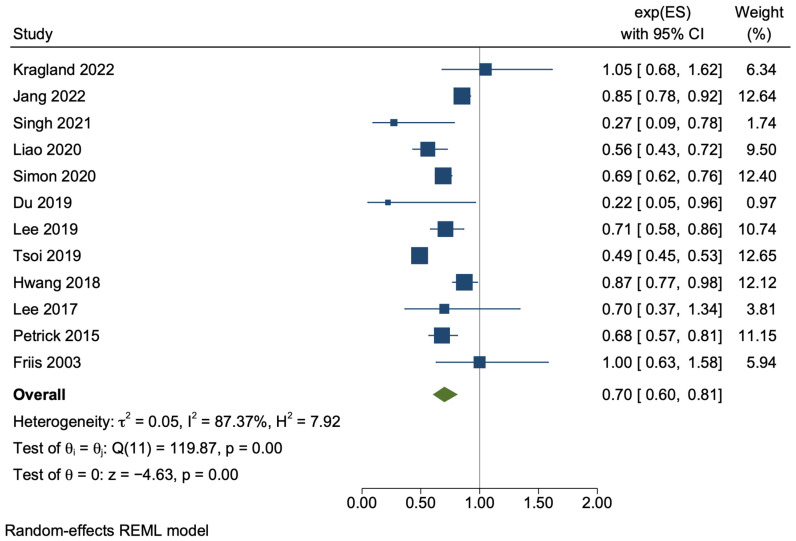
Random-effects meta-analysis of the pooled cohort studies [34,35,36,37,38,39,40,41,42,43,44,45].

**Figure 3 cancers-15-03518-f003:**
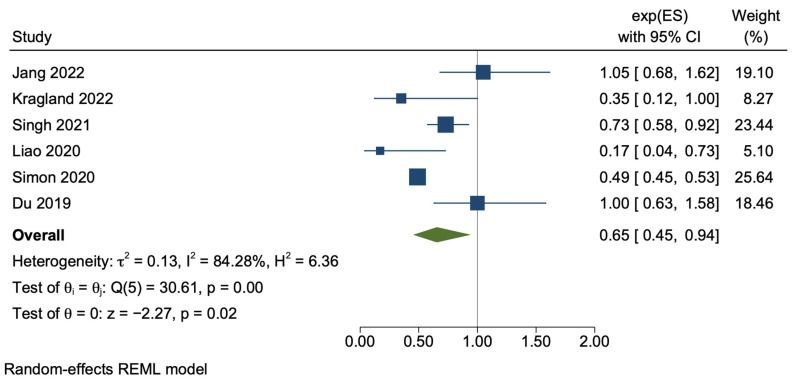
Random-effects meta-analysis of the pooled cohort studies, utilising only unadjusted data [34,37,38,41,43,44].

**Figure 4 cancers-15-03518-f004:**
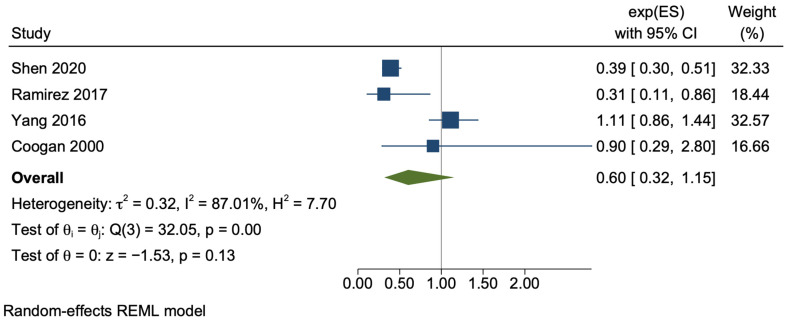
Random-effects meta-analysis of the pooled case-control studies [46,47,48,49].

**Figure 5 cancers-15-03518-f005:**
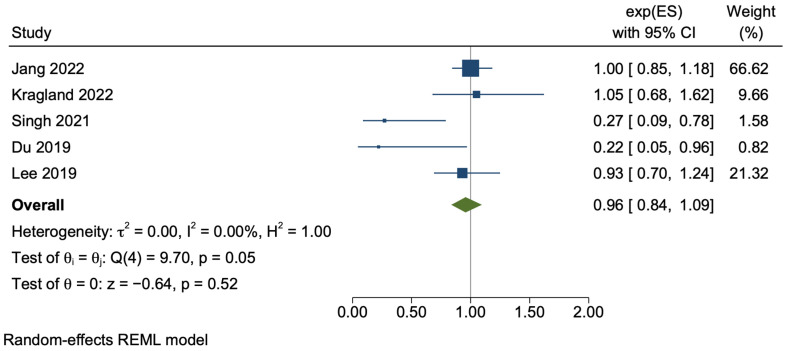
Subgroup random-effects meta-analysis of the patients with cirrhosis [34,37,38,39,44].

**Figure 6 cancers-15-03518-f006:**
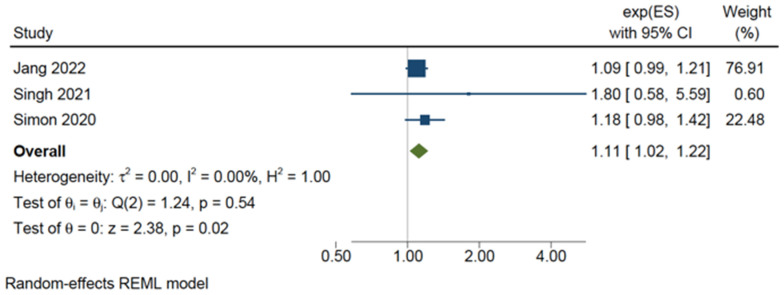
Random-effects meta-analysis of bleeding risk [37,43,44].

**Table 1 cancers-15-03518-t001:** Summary of eligible studies with overlapping study populations.

Population	Study	Years	Subjects (n)	Outcome
Hong Kong	Tsoi K 2019 [45]	2000–2004	612,569	Included
	Hui V 2021 [50]	2000–2018	35,111	Excluded
	Sung JJY 2020 [51]	2000–2004	405,533	
South Korea	Hwang C 2018 [36]	2002–2006	460,775	Included
	Jang H 2022 [37]	2007–2017	329,635	
	Jo JVH 2021 [52]	2005–2015	507,239	Excluded
	Yun B 2022 [53]	2010–2011	161,673	
	Lee M 2017 [54]	2002–2015	1674	
	Shin S 2020 [55]	2003–2016	949	
	Goh M 2020 [56]	2008–2012	7713	
	Cho K 2019 [57]	2003–2013	4980	
	Choi W 2021 [58]	2005–2015	32,695	
	Kim G 2017 [59]	2002–2013	1374	
Sweden	Simon T 2020 [43]	2005–2015	50,275	Included
	Brusselaers N 2018 [60]	2005–2012	NA ^a^	Excluded
Taiwan	Lee TY 2017 * [40]	1998–2012	18,080	Included
	Lee TY 2019 ** [39]	1997–2012	10,615	
	Liao Y 2020 *** [41]	2000–2012	3822	
	Chen VCH 2017 [61]	1997–2008	294,234	Excluded
	Chiu H 2010 [62]	2005–2008	1166	
	Wang CH 2022 [63]	2001–2011	3,008,665	
	Ho CM 2018 [64]	2005–2014	15,597	
	Huang HY 2022 [65]	1998–2000	5308	
	Lin Y 2018 [66]	2000–2011	18,243	
	Tseng C 2017 [67]	1999–2005	43,800	
USA	Petrick J 2015 ^i^ [42]	1980–2015	803,248	Included
	Singh J 2021 ^ii^ [44]	2012–2017	521	
	Coogan P 2000 ^ii^ [46]	1977–1998	5884	
	Ramirez A 2017 ^ii^ [47]	2012–2014	84	
	Shen Y 2020 ^ii^ [48]	2011–2016	1839	
	Sahasrabuddhe V 2012 ^iii^ [68]	1996–2008	300,504	Excluded
	Simon T 2018 ^iv^ [69]	1980–2010	133,371	

* Included patients with NAFLD; excluded anyone with a diagnosis of HBV or HCV. ** Included patients with HBV; excluded anyone with diagnosis of NAFLD or HCV. *** Included patients with HCV; excluded anyone with diagnosis of NAFLD or HBV. ^a^ Background population incidence rate rather than defined control group used. ^i^ Pooled analysis of multiple US cohort studies (NIH-AARP Diet and Health Study (AARP), Agriculture Health Study (AHS), United State Radiologic Technologist Study (USRT), The Breast Cancer Detection Demonstration Project (BCDDP), Prostate, Lung, Colorectal and Ovarian Cancer Screening Trial (PLCO), Health Professionals Follow-up Study (HPFS), Cancer Prevention Study II (CPSII), Black Women’s Health Study (BWHS), Women’s Health Initiative (WHI), Nurses’ Health Study (NHS)). ^ii^ Local studies. ^iii^ NIH-AARP Diet and Health Study (AARP). ^iv^ Pooled analysis of two US cohort studies (Health Professionals Follow-up Study (HPFS), Nurses’ Health Study (NHS)).

**Table 2 cancers-15-03518-t002:** The baseline characteristics and bias assessment of the included cohort studies.

Author	Year	Country	Population	Cohort (n)	Aspirin Users (n)	HCC Cases (n)	mNOS
Du [34]	2019	China	Viral cirrhosis	264	59	41	9
Friis S [35]	2003	Denmark	Public		29,470	21 *	6
Hwang C [36]	2018	South Korea	Public	460,755	64,782	2336	9
Jang H [37]	2022	South Korea	HBV	329,635	20,200	2697	9
Kraglund F [38]	2023	Denmark	Alcohol cirrhosis	115,092	1449	2830	9
Lee TY. [40]	2017	Taiwan	NAFLD	18,080	5602	41	7
Lee TY [39]	2019	Taiwan	HBV	10,615	2123	697	8
Liao Y [41]	2020	Taiwan	HCV	3822	2980	278	9
Petrick J [42]	2015	USA	Public	803,248	477,470	679	7
Simon T [43]	2020	Sweden	HBV/HCV	50,275	14,205	1612	9
Singh J [44]	2021	USA	Cirrhosis	521	170	45	9
Tsoi K [45]	2019	Hong Kong	Public	612,569	204,170	9370	8

* Number of HCC cases in aspirin users.

**Table 3 cancers-15-03518-t003:** The baseline characteristics and bias assessment of the included case-control studies.

Author	Year	Country	Population	Cases (n)	Controls (n)	mNOS
Coogan [46]	2000	USA	Public	51	5952	6
Ramirez [47]	2017	USA	Public	42	42	8
Shen [48]	2020	USA	Public	673	1166	9
Yang [49]	2016	UK	Public	1195	4640	7

## Data Availability

All relevant data are included within the manuscript and Appendix A.

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
