# Peer review of "The Effect of Aspirin Use on Incident Hepatocellular Carcinoma—An Updated Systematic Review and Meta-Analysis"

_cancers, 2023, doi:10.3390/cancers15133518_

Round 1
Reviewer 1 Report
This is a beautifully narrated article. I have a few queries
1. Rationale behind the selection of Aspirin needs much justification in the abstract and the content
2. The inclusion of articles is very few. why that many articles need an exclusion?
3. One more figure is needed
4. The content needs clarity in the results and discussion part
5. Discussion needs clarity why they need such an analysis.. compare with the existing ones.
6 Please look into the following article
Hepatology Communications 2021;5:133-143 How it differs form your article
7. have you gone through the existing similar kind of articles? That need to be mentioned in the PRISMA anaysis
The manuscript needs substantial revision with aforementioned comments
Author Response
We thank the reviewer for the helpful comments, feedback and suggestions and respond to the critiques as follows:
Response to Reviewer 1
- Rationale behind the selection of Aspirin needs much justification in the abstract and the content.
- We have added the following sentence in the abstract: ‘An increasing number of observational studies have described an association between aspirin use and reduced risk of incident hepatocellular carcinoma.‘
- Please note we have endeavoured to explain the rationale for the selection of aspirin assessment in the Introduction on Page 2 Para.2 as follows: ‘Aspirin is an inexpensive widely available drug that targets the cyclo-oxygenase group of enzymes. It is thought, based on animal studies, to interfere with HCC carcinogenesis through both its anti-platelet(11, 12) and anti-inflammatory(13, 14) The role of aspirin in primary prevention of colorectal cancer has been definitively established(15-17), however its utility in prevention of other cancers remains subject to investigation. Several observational studies have been published to date attempting to assess the effect of aspirin use on incident HCC. The results have varied, however, recently a number of large high-quality studies have suggested that there is a duration-dependent effect of aspirin in reducing the incidence of HCC supporting a causal relationship.’
- We have added a paragraph added to the discussion: ‘Aspirin is widely used in the primary and secondary prevention of cardiovascular disease however its anti-cancer effects (primarily through its anti-cyclooxygenase activity) have been widely described, including in liver cancer mouse models(11, 14). It is now recommended for consideration as a preventative treatment in colorectal cancer, with a broad evidence base of observational and randomized controlled trials supporting its efficacy(15, 72-78). To date, there have been a large number of observational studies reporting a negative association between aspirin use and HCC but no randomized controlled trials have been performed yet.’
- Also note this element in the discussion: ‘Given the postulated mechanism of aspirin’s chemoprophylactic role in hepatocarcinogenesis is primarily through a reduction in hepatic accumulation of inflammatory cells and reduction in the hepatocellular injury that drives oncogenic mutations(12), it is plausible that this effect would have a diminished impact in the already cirrhotic population and maximal impact in earlier stages of liver disease.’
- The inclusion of articles is very few. why that many articles need an exclusion?
In order to avoid bias and an overestimation of the aspirin effect via the inclusion of duplicate subjects and HCC cases, we excluded studies where there was unequivocal overlapping of study populations. Because of the large number of studies were clearly drawing from the same cohorts (Taiwan National Health Insurance Research Database, the South Korea National Health Insurance Service Database, the Hong Kong Hospital Authority Clinical Data repository in particular), a large number of studies needed to be excluded.
- We have explained this in the discussion with greater clarity with the following - ‘Multiple meta-analyses(18-30) have been published over the last few years summarizing the observational data in this area, with the majority of these concluding that aspirin does have a significant association with reduced HCC risk. Unfortunately, the majority of these systematic reviews have introduced bias in their risk estimates via the erroneous inclusion of multiple studies drawing from the same population. By mistakenly including the same subjects and HCC cases more than once, the estimate of precision of these analyses will have been overestimated and depending on which populations were duplicated, the overall risk estimate may have been significantly swayed. Due to concern for the unreliability of previous meta-analyses on this basis, as well as the interval publication of further high quality studies(37, 38), we sought to perform an updated and methodologically-sound meta-analysis in order to provide the most accurate and updated insight into the effect of aspirin use on HCC incidence.’
- We have outlined our approach in the method section 2.3: ‘In cases of multiple studies drawing from the same population registry, the years of data extraction and exclusion criteria were noted. Studies were carefully chosen with preference given to the highest quality study as assessed by modified Newcastle-Ottawa Quality Assessment Scale (mNOS)(33) and if there was equal quality, the study or studies involving the largest number of subjects possible was selected.’
- We describe the need to exclude a large number of studies in method section 3.1 ‘A large number of cohort studies involved the same population, in part due to the availability of large national databases such as the Taiwan National Health Insurance Research Database, the South Korea National Health Insurance Service Database and the Hong Kong Hospital Authority Clinical Data repository - as well as duplication of efforts in the same longitudinal US cohorts and European population registries. A total of 20 studies were excluded due to overlapping study populations (8 South Korea, 7 Taiwan, 2 USA, 2 Hong Kong, 1 Sweden) and the details of these are outlined in Table 1.’
- We have outlined in Table 1 all of the overlapping studies, based on common population registry and overlapping years of screening – and which were included/excluded with specific subtleties outlined in the footer for the Taiwan and US studies
- One more figure is needed
- We have changed the ‘Figure S1: Random-effects meta-analysis of pooled cohort studies utilizing only crude unadjusted date’ to a main figure in the body of the text (Figure 3)
- There are now a total of six figures and only two supplementary figures – labels have been adjusted accordingly.
- The content needs clarity in the results and discussion part
- We have addressed this issue as noted above.
- Discussion needs clarity why they need such an analysis. Compare with the existing ones.
We have added the following to the third paragraph of the discussion ‘Multiple meta-analyses(18-30) have been published over the last few years summarizing the observational data in this area, with the majority of these concluding that aspirin does have a significant association with reduced HCC risk. Unfortunately, the majority of these systematic reviews have introduced bias in their risk estimates via the erroneous inclusion of multiple studies drawing from the same population. By mistakenly including the same subjects and HCC cases more than once, the estimate of precision of these analyses will have been overestimated and depending on which populations were duplicated, the overall risk estimate may have been significantly swayed. Due to concern for the unreliability of previous meta-analyses on this basis, as well as the interval publication of further high quality studies(37, 38), we sought to perform an updated and methodologically-sound meta-analysis in order to provide the most accurate and updated insight into the effect of aspirin use on HCC incidence.’
In the fourth paragraph of our discussion we draw attention to the difference in findings of our meta analysis compared with those previously published – ‘However, unlike previous reports, our meta-analysis is the first to report a diminished risk reduction effect in cirrhosis and is also the first to observe, in this setting, a statistically significant increase in bleeding events associated with aspirin.’
- Please look into the following article - Hepatology Communications 2021;5:133-143 How it differs from your article
This paper has been cited already in our paper (Reference 23) together with an additional 12 other meta analyses
We have edited our discussion to include the following ‘Multiple meta-analyses(18-30) have been published over the last few years summarizing the observational data in this area, with the majority of these concluding that aspirin does have a significant association with reduced HCC risk. Unfortunately, the majority of these systematic reviews have introduced bias in their risk estimates via the erroneous inclusion of multiple studies drawing from the same population. By mistakenly including the same subjects and HCC cases more than once, the estimate of precision of these analyses will have been overestimated and depending on which populations were duplicated, the overall risk estimate may have been significantly swayed. Due to concern for the unreliability of previous meta-analyses on this basis, as well as the interval publication of further high quality studies(37, 38), we sought to perform an updated and methodologically-sound meta-analysis in order to provide the most accurate and updated insight into the effect of aspirin use on HCC incidence.’
- Differences between our systematic review and Memel et al., Hepatology Communications 2021;5:133-143 include:
- Memel’s review included studies with overlapping study populations (which introduces bias) that we excluded.
- Hwang 2018 has overlap with Shin 2020, Lee M 2017, Kim 2017
- Petrick 2015 has overlap with Saharabuddhe 2012
- Tseng 2018 has overlap with Lee T 2017
- Lin 2018 has overlap with Lee T 2017
- Our review includes high quality studies published after Memel’s review
- Jang 2022, Kraglund 2023, Singh 2021
- By reducing bias through excluding overlapping study populations and by including new studies we have different conclusions to Memel’s review, namely:
- Memel reports a statistically significant HCC risk reduction in the cirrhosis subgroup while we found a diminished non significant risk reduction – this is a critical difference and highlights that the optimal patient group to benefit may be those with non-cirrhotic chronic liver diseases
- Memel reported a non-association between aspirin use and bleeding risk without performing a meta analysis while we performed a meta analysis and found a statistically significant increase in bleeding risk – which is of crucial practical importance in considering the risks and benefits of real world chemoprophylactic use of aspirin.
- Have you gone through the existing similar kind of articles? That need to be mentioned in the PRISMA analysis
Note in the initial manuscript the last sentence of section 2.1 ‘Reference lists of relevant manuscripts were also screened to identify further studies for review.’
- This has been changed to ‘Reference lists of relevant manuscripts were also screened to assess for potential studies missed from the systematic search results, however no such additional studies were identified.’

Reviewer 2 Report
HBeAg and HBV DNA are important factors related to the development of HCC. Antiviral therapy for HBV or HCV might decrease the risk of HCC development. Did all these studies include these data? If not, potential bias might influence the result. This point needs some discussion.
Author Response
We thank the reviewer for the helpful comments, feedback and suggestions and respond to the critique as follows:
Response to Reviewer 2
- HBeAg and HBV DNA are important factors related to the development of HCC. Antiviral therapy for HBV or HCV might decrease the risk of HCC development. Did all these studies include these data? If not, potential bias might influence the result. This point needs some discussion.
- We have changed this sentence in the last paragraph of the discussion ‘Similarly, there was variable adjustment for multiple other key factors known to affect HCC risk such as alcohol use, presence of NAFLD/metabolic risk factors, family history of HCC and viral status’ to
‘Status of the oncogenic hepatitis B virus was even less well-documented and adjusted for, with none of the case control studies and only 5 out of the 12 cohort studies accounting for hepatitis B status in their adjusted risk estimates (3 studies(37, 39, 43) by adjusting for antiviral use, 2 studies(40, 41) by excluding all chronic hepatitis B cases). It is certainly possible for adherence to aspirin therapy to be associated with adherence to antiviral medication thereby confounding the results in the studies that failed to control for this key factor.’
And ‘Similarly, there was generally poor documentation and adjustment for multiple other key factors known to affect HCC risk such as alcohol use, presence of NAFLD/metabolic risk factors, family history of HCC and hepatitis C and hepatitis delta viral status.’

Round 2
Reviewer 2 Report
The authors have revised the manuscript according to my previous comments.